# Study of Ion-to-Electron Transducing Layers for the Detection of Nitrate Ions Using FPSX(TDDAN)-Based Ion-Sensitive Electrodes

**DOI:** 10.3390/s24185994

**Published:** 2024-09-15

**Authors:** Camille Bene, Adrian Laborde, Morgan Légnani, Emmanuel Flahaut, Jérôme Launay, Pierre Temple-Boyer

**Affiliations:** 1CNRS, LAAS, 7 Avenue du Colonel Roche, F-31400 Toulouse, France; camille.bene@laas.fr (C.B.); adrian.laborde@laas.fr (A.L.); launay@laas.fr (J.L.); 2INSAT, UT3-PS, INP, University of Toulouse, 118 Route de Narbonne, CEDEX 9, 31062 Toulouse, France; 3Toulouse INP, CNRS, CIRIMAT, Université Toulouse 3 Paul Sabatier, Toulouse INP, CNRS, Université de Toulouse, 118 Route de Narbonne, CEDEX 9, 31062 Toulouse, France; morgan.legnani@univ-tlse3.fr (M.L.); emmanuel.flahaut@univ-tlse3.fr (E.F.)

**Keywords:** ion-sensitive electrode, potentiometric sensor, ElecCell device, ion-sensitive layers, nitrate NO_3_^−^ ion, water analysis

## Abstract

The development of ISE-based sensors for the analysis of nitrates in liquid phase is described in this work. Focusing on the tetradodecylammonium nitrate (TDDAN) ion exchanger as well as on fluoropolysiloxane (FPSX) polymer-based layers, electrodeposited matrixes containing double-walled carbon nanotubes (DWCNTs), embedded in either polyethylenedioxythiophene (PEDOT) or polypyrrole (PPy) polymers, ensured improved ion-to-electron transducing layers for NO_3_^−^ detection. Thus, FPSX-based pNO_3_-ElecCell microsensors exhibited good detection properties (sensitivity up to 55 mV/pX for NO_3_ values ranging from 1 to 5) and acceptable selectivity in the presence of the main interferent anions (Cl^−^, HCO_3_^−^, and SO_4_^2−^). Focusing on the temporal drift bottleneck, mixed results were obtained. On the one hand, relatively stable measurements and low temporal drifts (~1.5 mV/day) were evidenced on several days. On the other hand, the pNO_3_ sensor properties were degraded in the long term, being finally characterized by high response times, low detection sensitivities, and important measurement instabilities. These phenomena were related to the formation of some thin water-based layers at the polymer–metal interface, as well as the physicochemical properties of the TDDAN ion exchanger in the FPSX matrix. However, the improvements obtained thanks to DWCNT-based ion-to-electron transducing layers pave the way for the long-term analysis of NO_3_^−^ ions in real water-based solutions.

## 1. Introduction

Nitrate ions are essential in the nitrogen cycle and therefore in the growth and development of plants. In this frame, plants assimilate nitrogen in the form of nitrate (NO_3_^−^), nitrite (NO_2_^−^), and ammonium (NH_4_^+^) ions, and transform it into organic matter through photosynthesis [1]. The final decomposition of this organic matter leads to the release of ammonia (NH_3_), which is then oxidized through bacterial activity, producing nitrate compounds in soils (concentration range: 1–5 mM) [2].

As a matter of fact, since nitrate ions play a main role in the soil–plant system [3], the development of nitrogen-based fertilization has become essential for improving agricultural production in terms of quantity and quality. Nevertheless, the excessive use of nitrogen-based fertilizers in modern farming leads today to a worldwide disruption of the nitrogen cycle. In excess, nitrate ions are finally responsible for the degradation of ecosystems through soil leaching, eutrophication of fresh and marine waters, pollution of groundwaters and drinking waters, emission of ammoniac (NH_3_) and nitrogen oxide (NO_x_) gases, etc. [4,5,6]. They are also known to be ultimately dangerous to health if found in drinking water, as NO_3_^−^/NO_2_^−^ ions are likely to form nitrogen compounds suspected of being reprotoxic and carcinogenic [7]. It has therefore become essential to focus on nitrate detection in the frame of environmental analysis [8,9]. Different analysis techniques have been developed, with the emphasis on electrochemical ones, since they allow the development of low-cost, integrated, potentiometric pNO_3_ sensors using different microtechnological platforms based on ion-sensitive field effect transistors (ISFETs) [10,11,12,13,14,15,16,17] or ion-sensitive electrodes (ISEs) [18,19,20,21,22,23,24,25,26,27].

In the frame of the ISFET platform, since the use of an electrolyte–insulator–semiconductor (EIS) structure prevents the integration of an ion-to-electron transducing layer in principle, NO_3_^−^-sensitive membranes were developed using varied polymers and associated ionophores or ion exchangers (Table 1). However, on the basis on our last results [17], if tetradodecylammonium nitrate (TDDAN) was found to be well-adapted to fluoropo-lysiloxane-based ion-sensitive membranes, our FPSX-pNO_3_-ISFET devices were still characterized by measurement instabilities in water-based solutions, allowing some possible improvements thanks to specific interface layers.

In the frame of the ISE platform, such instabilities are more problematic due to the use of a conductive electrolyte–metal interface. They should be dealt with more carefully and, for this, ion-to-electron transduction layers are usually required [18,19,20,21,22,23,24,25,26,27]. As a result, ion-to-electron transducers as well as ion-sensitive layers were studied simultaneously in order to further improve pNO_3_ measurement properties in liquid phase (Table 2).

In previous studies dedicated to the ISFET technological platform [17], a process was optimized in order to integrate FPSX-based ion-sensitive layers (thickness: ≈6 µm) on a silicon nitride Si_3_N_4_ pH-sensitive layer. In the frame of the ElecCell technological platform, even if an SiN_x_ upper passivation layer allows good adhesion properties on the different electrodes, ionic measurement properties of the corresponding electrolyte–insulator–solid (EIS) structure are known to be degraded by long-term immersion in liquid phase. Apart from the ion-sensitive or ion-exchanging properties, ISE theoretical studies relate such degradations to the formation of a thin water layer between the insulative polymeric membrane and the metallic electrode and to the corresponding charge transfer alteration. In the frame of an FPSX-pNO_3_-ElecCell technological platform, and in order to cope with such water-based instabilities, it therefore seems appropriate to integrate an ion-to-electron transducing layer (IETL), aiming at the improvement of the electrochemical transduction between the FPSX-based ion-sensitive layer and the platinum-based working microelectrode.

As a matter of fact, this paper proposes such studies, dealing with the use of double-walled carbon nanotubes (DWCNTs) as well as polymer-based layers, i.e., polyethylenedioxythiophene (PEDOT) and polypyrrole (PPy), and aiming at the long-term improvement of NO_3_^−^ ion-sensitive layers for liquid-phase analysis.

## 2. Materials and Methods

### 2.1. Microdevice Fabrication

Silicon technologies were used in order to mass integrate an electrochemical microcell (ElecCell; Figure 1), according to microfabrication processes previously studied [28]. Starting from an oxidized silicon wafer (SiO_2_ thickness: ≈600 nm), different thin-film metallic layers (Ti: ~20 nm, Pt: ~200 nm, and Ag: ~400 nm) were deposited by physical vapour deposition and patterned by photolithography, leading to the integration of a platinum-based 5 × 5 ultramicroelectrode array (UMEA; microelectrode radius: ~5 µm, total active area: ~2000 µm^2^) used as a working electrode, a platinum-based counter microelectrode, as well as a silver-based reference microelectrode. Finally, a silicon nitride (SiN_x_) wafer-level passivation (thickness: ≈200 nm) was performed using a low-temperature PVD process in order to define precisely the microelectrode surface and to ensure long-term electrochemical stability [29]. Although the UMEA structure may be less optimal, since the metal–membrane surface is reduced [30], it was finally selected since the large SiN_x_ surface allows better physicochemical adhesion of the FPSX-based membrane on the platinum-based layer.

The (Pt - Pt - Ag) ElecCell devices were manufactured on 5 × 6 mm^2^ chips. These chips were stuck on specifically coated printed circuit boards using an epoxy insulating glue. After wire bonding, packaging was performed at the system level using a silicone glop-top in order to adapt the final device to the detection in liquid phase (Figure 2). The final pseudo-reference Ag/AgCl was formed through electrochemical oxidation of silver in potassium chloride (10^−1^ M) by linear sweep voltammetry from 0.1 to 0.25 V vs. SCE with a scan rate of 1 mV·s^−1^. This process allows the obtainment of a thin, homogeneous, and chemically stable AgCl film, ensuring a temporal drift of 0.4 mV/h in a one-day period.

The three-electrode ElecCell design was chosen for its suitability for both amperometry and potentiometry detection techniques in liquid phase, aiming for future environmental applications. Dealing with potentiometry, the platinum counter microelectrode is unnecessary, limiting studies to a two-electrode system. Furthermore, in the frame of the current work, a commercial reference electrode (see below) was finally used in order to prevent any interfering measurement phenomena associated with the silver–silver chloride pseudo-reference microelectrode (temporal drift, ageing, etc.).

### 2.2. Integration of Ion-Sensitive Membranes in ElecCell Devices

Ion detection was investigated using fluoropolysiloxane polymer (FPSX 730 FS, purchased from Dow Corning), known to present improved properties for the integration of polymer-based ion-sensitive layers [31,32]. Except for this FPSX polymer, all the other chemical reagents (see below) were purchased from Sigma-Aldrich.

The initial FPSX-based polymeric solution was made of 200 mg of fluoropolysiloxane centrifuged in 1.5 mL of tetrahydrofuran. Thus, according to previous studies [17], the NO_3_^−^ ion-sensitive solution was obtained by mixing 4.2 mg of tetradodecylammonium nitrate (TDDAN) used as an ion exchanger; 2.5 mg of potassium tetrakis [3,5-bis(trifluoromethyl)phenyl] borate (KTFBP) used as an ionic additive, i.e., for a TDDAN:KTFPB molar ratio of 2:1; and 93.3 mg of the FPSX-based solution.

Before depositing the FPSX-based ion-sensitive layers, different ion-to-electron transducing matrixes were studied in order to improve the electronic transduction with the platinum-based metallic layer [23,25,33,34,35,36]:-Matrix #1: poly (3,4-ethylenedioxythiophene) (PEDOT) doped with sodium polystyrene sulfonate (NaPSS);-Matrix #2: poly (3,4-ethylenedioxythiophene) (PEDOT) doped with double-walled carbon nanotubes (DWCNTs);-Matrix #3: polypyrrole (PPy) doped with double-walled carbon nanotubes (DWCNTs).

In the following, all the electrodeposited solutions were stored in the dark at 4 °C. Furthermore, using a VMP3 potentiometer from Biologic, all the electropolymerization/ deposition processes were conducted thanks to the integrated platinum-based counter microelectrode in order to ensure the best deposition homogeneity possible. In each case, electrochemical cleaning of the platinum-based ultramicroelectrode array was first carried out by cycling voltammetry between −0.2 and 1.4 V versus Ag/AgCl/KCl_saturated_ at a 200 mV·s^−1^ scan rate in a 0.5 M solution of sulfuric acid H_2_SO_4_. Fifty cycles were found to be efficient in order to remove organic impurities from the platinum surface.

Matrix #1 was prepared in two steps. At first, 0.35 g of NaPSS was dissolved in 50 mL deionized water and stirred for 12 h. Then, 54.54 µL EDOT was added, and the final solution was stirred again for 2 h to ensure a good homogeneity. Finally, the electrodeposition process was performed by chronopotentiometry at a fixed current of 10 pA·µm^−2^ current density for 6 min.

For Matrixes #2 and #3, the synthesis of double-walled carbon nanotubes (DWCNTs) was described elsewhere [36]. Immediately after synthesis, they were oxidized in nitric acid (HNO_3_, 3 mol·L^−1^) for 24 h in reflux conditions at 130 °C. Then, the oxidized nanotubes were filtered and washed with deionized water until pH neutrality. Oxidized DWCNTs were kept humid so that they could be easily redispersed. Finally, the DWCNT suspension was prepared by diluting wet oxidized carbon nanotubes at 2 mg·mL^−1^ in deionized water.

Concerning the second matrix [36], the initial solution was prepared by adding 10.9 µL EDOT in 5 mL deionized water. After a two-hour stirring, 5 mL of the prepared DWCNT suspension, previously ultrasonicated for half an hour, was added. The full mixture was again stirred (duration: 2 h) and sonicated (duration: 0.5 h) to ensure sufficient homogeneity. The electrodeposition process was finally conducted by chronopotentiometry at a 10 pA·µm^−2^ current density for 6 min.

The third matrix was prepared in a similar way [36], ensuring good dissolution and homogeneity properties. A quantity of 75 µL of pyrrole was dissolved in deionized water and stirred for 15 min. Then, 5 mL of the DWCNT sonicated solution was added, and the complete mixture was again stirred and sonicated (see below). The final ion-to-electron transducing layer was electrodeposited by chronopotentiometry at a 10 pA.µm^−2^ current density for 8 min.

After the electrodeposition of the ion-to-electron transducing layer (IETL), the FPSX-based ion-sensitive layer was deposited on top of the UMEA-based working microelectrode. In all cases, the FPSX-based ion-sensitive solutions (see below) were mixed in an ultrasonic bath (duration: 30 min) to ensure homogeneity and then deposited by drop casting using a Hamilton micro syringe (volume: 5 µL). Finally, THF evaporation and FPSX reticulation were performed at ambient temperature and atmospheric moisture. Thus, quasi-semi-ellipsoidal (radius: 1250 ± 50 µm, height: 12 ± 1 µm) FPSX(TDDAN)-based membranes were deposited with dimensions compatible with the platinum-based square (area: 2340 × 2340 µm^2^) associated with the ultramicroelectrode array (Figure 3). Finally, the working microelectrode was adapted to NO_3_^−^ detection, leading to the realisation of an Pt/IETL/FPSX(TDDAN) NO_3_^−^-sensitive structure (Figure 4) as well as the associated UMEA-based (Pt/IETL/FPSX(TDDAN) − Pt − Ag/AgCl) ElecCell (called pNO_3_-ElecCell hereafter).

### 2.3. Physical Characterization of Ion-Sensitive Membranes

Optical microscopy and scanning electron microscopy (SEM) were respectively performed using a numerical microscope, the HRX-01 from HIROX, and a dual-beam FIB/SEM HELIOS 600i from FEI. After a cut was made with the focused ion beam (FIB), this last piece of equipment was also used to analyse the main membrane properties (thickness, roughness, porosity, and density). Finally, ion-sensitive membrane thicknesses were also evaluated by profilometry using a 3D optical profilometer, the SNEOX from Sensofar.

### 2.4. Electrochemical Characterization in Liquid Phase

Using an Ag/AgCl/KCl_saturated_ reference electrode from SI Analytics as well as a VMP3 potentiometer from Biologic, the different pNO_3_-ElecCell devices were characterized through open-circuit voltage measurements in KNO_3_ solutions for concentrations ranging between 10^−6^ and 10^−1^ M, aiming at the study of their detection properties (sensitivity, measurement range, and stability).

Concerning selectivity, three anions associated with environmental analysis, i.e., chloride (Cl^−^), hydrogen carbonate (HCO_3_^−^), and sulfate (SO_4_^2−^) ions, were also studied thanks to the fixed interference method (FIM; [37]) for a concentration of interfering ions of 10^−2^ M.

Finally, all chemical products were purchased from Sigma-Aldrich, and all experiments were performed at room temperature (~21 °C).

## 3. Results and Discussion

### 3.1. Physical Analysis of the Different Polymer-Based Membranes

The morphology of the PEDOT-based and PPy-based transducing layers was studied by optical microscopy, evidencing different configurations and structures (Figure 5 and Figure 6). In this context, it should first be mentioned that, in the frame of the SiN_x_ wafer-level passivation process, the ultramicroelectrode (UME) radius was reduced, decreasing from its theoretical value (5 µm) to its technological one (~4 µm) [29].

The PEDOT:PSS electrodeposition process (Matrix #1) was found to be reproducible, allowing a homogeneous deposition on all the 25 interconnected platinum ultramicroelectrodes (Figure 5a and Figure 6a). The PEDOT-based UME radius was finally estimated at around 5.5 µm, evidencing a conformal deposit. On the contrary, concerning the different DWCNT-rich layers (Matrixes #2 and #3), due to higher electrochemical kinetics, layers were deposited largely beyond the electroactive surface (Figure 5b,c). Consequently, the modified UME radii were greatly increased, reaching 16.5 µm and 12 µm for the PEDOT-based and PSS-based electrodeposition processes, respectively. In fact, only the PPy:DWCNT electrodeposition process was effectively reproducible, allowing a homogeneous deposition on all the 25 interconnected platinum ultramicroelectrodes, even if some CNT-related defects were still evidenced (Figure 6c). On the contrary, the PEDOT:DWCNT one was hardly reproducible, leading to the formation of CNT-related clusters and to higher roughness (Figure 6b). In this case, apart from the presence of carbon nanotubes, such phenomena should also be related to the low solubility of the EDOT monomer in NaPSS-free aqueous solutions [34,38,39].

### 3.2. pNO_3_-ELecCell Characterization

The NO_3_^−^ detection was studied while focusing on concentration ranges compatible with environmental analysis, i.e., [10^−6^–10^−1^ M]. According to the IETL integration, four different pNO_3_-ElecCell devices were studied (Figure 7).

In the absence of transducing layers, degraded performances were obtained when compared to FPSX(TDDAN)-pNO_3_-ISFET [17], evidencing a low detection sensitivity (~30 mV/pNO_3_), a low detection range, and high response times (Figure 7a). In the frame of the ion-sensitive electrode (ISE) study, such degradations are known to be related to the formation of a thin water layer between the insulative polymeric membrane and the metallic electrode and therefore to measurement instabilities of the whole electrolyte–insulator–solid (EIS) interface. In order to cope with such water-based instabilities, ion-to-electron transduction was considered when dealing with FPSX(TDDAN)-pNO_3_-ElecCell. Thus, whatever the transducing layer, improved performances were obtained for the different devices, from the least to the best (Figure 7b–d and Figure 8, Table 3).

First, considering the PEDOT:PSS transducing layer (Matrix #1), if the response time was faster, the detection sensitivity was always around ~30 mV/pNO_3_—far lower than the Nernstian law. Since the PEDOT electropolymerization/deposition process was performed using water-based solutions, such results should effectively be related to some thin water-based layers at the polymer–metal interface.

As the use of carbon nanotubes was previously proposed in order to increase double-layer capacitance [23,25,30,40,41], double-walled carbon nanotubes (DWCNTs) were added to the polymer-based electrodeposited solution, i.e., polyethylenedioxythiophene and polypyrrole (Matrixes #2 and #3, respectively). As a result, detection performances were considerably improved, evidencing very low time constants (<1 s) and quasi-Nernstian sensitivities (50–55 mV/pNO_3_) for both the PEDOT:DWCNT and PPy:DWCNT interfaces. Compared to the Nernst law, such lower detection sensitivities are typical of a non-ideal equilibrium at the electrolyte–membrane interface. They should be related to the physicochemical NO_3_^−^-based ion-exchanging properties of the TDDAN molecule in the FPSX structure.

As a matter of fact, the best analytical results were obtained for the PEDOT:DWCNT matrix. Nevertheless, taking into account the non-reproducibility of the associated deposition process, the best compromise was finally associated with the PPy:DWCNT one.

Focusing on these DWCNT-based matrixes, the fixed interference method (FIM) was used to determine the potentiometric selectivity coefficients for the two different pNO_3_-ElecCell devices, dealing with three anions associated with environmental analysis: chloride (Cl^−^), hydrogen carbonate (HCO_3_^−^), and sulphate (SO_4_^2−^). Thus, very similar experimental results were obtained for the PEDOT:DWCNT and PPy:DWCNT matrixes, even if the hydrogen carbonate was found to be responsible for a strong pNO_3_-sensitivity decrease in the second one (Figure 9). Since the HCO_3_^−^ ion is associated with two weak acid–base couples (H_2_CO_3_/HCO_3_^−^: pKa_1_ = 6.37 and HCO_3_^−^/CO_3_^2−^: pKa_2_ = 10.32), this lower sensitivity (~30 mV/pNO_3_) should be related either to the protonation of the pyrrole molecule and therefore to the PPy sensitivity to pH [42,43] or to the sensitivity of PPy to dissolved carbon dioxide (CO_2_) in liquid phase and therefore to the carbonic acid (H_2_CO_3_) concentration in solution [44], both phenomena being undoubtedly linked.

Thus, low selectivity coefficients were obtained for both matrixes, in global agreement with the Hofmeister series of anions [45]. Such results remain quite good, since values below −2 should be enough in the frame of environmental analysis.

Finally, the measurement stability of the two different DWCNT-based pNO_3_-ElecCell devices was studied. A first experiment was performed in a 10^−2^ M KNO_3_ solution. In such conditions, both sensors showed a low voltage drift (~1.5 mV/day) over two–three days, followed by a slow but certain degradation over time. Concerning the initial middle-term drift, such a result is a decade higher than those described in the literature, i.e., ≈60 µV/h versus ≈6 µV/h [20,21,26], but could be further improved by developing a specific packaging procedure.

Nevertheless, since the main bottleneck of pNO_3_-ISE is associated with its long-term stability as well as reusability, an experimental campaign was also planned to check the pNO_3_ analytical response after one month of dry storage (Table 4). If a negligible shift was evidenced after one day, this was no longer for longer periods. After one week, measurement instabilities and drifts were evidenced for both devices in the [1,2,3,4,5] pNO_3_ range, resulting in a low increase in measurement error and a slight decrease in sensitivity. Nevertheless, such results demonstrate that the developed pNO3-ISE is quite reusable after dry storage.

After one month, the situation was greatly worsened, especially for the Pt/PPy:DWCNT/FPSX(TDDAN) matrix: the pNO_3_ sensitivities dropped dramatically, while the associated time constants were multiplied by around one hundred. Such results had already been shown for pNO_3_-ISFET devices and were related to the physicochemical properties of the TDDAN ion exchanger [17]. Nevertheless, according to use in water-based solutions, ageing, swelling, and detachment phenomena of the whole IETL/FPSX ion-sensitive membranes were not overlooked. The temporal degradation of the transducing layer properties was obviously responsible for the final appearance of some thin water-based layers at the polymer–metal interface. As a matter of fact, in the frame of FPSX(TDDAN)-pNO_3_-ElecCell/ISE devices, the study of DWCNT-based ion-to-electron transducing layers remains promising, since stable measurements and quasi-Nernstian responses were obtained for up to one week, paving the way for NO_3_^−^ detection in the frame of environmental analysis.

## 4. Conclusions

Silicon-based technologies were used to develop ElecCell-based sensors for water analysis, dealing with the TDDAN ion exchanger associated with nitrate (NO_3_^−^) ions, as well as fluoropolysiloxane (FPSX) based ion-sensitive layers. Thus, the use of DWNCNT-based ion-to-electron transducing layers was found to be mandatory, and PEDOT:DWCNTs as well as PPy:DWCNTs gave promising results for NO_3_^−^ detection in liquid phase while considering the integration process, detection sensitivity and selectivity, response time, middle-term stability, and reusability.

As a matter of fact, the FPSX(TDDAN) pNO_3_-ElecCell devices were characterized by quasi-Nernstian detection properties (sensitivity higher than 50 mV/decade) in the [10^−1^–10^−5^ M] concentration range, in agreement with detection specifications related to freshwater analysis. Their corresponding selectivity properties were also studied for different water-based interfering anions, evidencing low potentiometric selectivity coefficients in all cases (log K < −2).

Focusing on the measurement instabilities associated with the TDDAN molecule, improved results were obtained, since a low temporal drift (~1.5 mV/day) was evidenced after a fortnight’s duration, evidencing a real improvement in the frame of long-term applications. Nevertheless, after a one-month duration, due to the final appearance of a thin water-based layer at the polymer–metal interface and to the temporal degradation of the transducing layer, the pNO_3_ sensor properties were characterized by measurement instabilities responsible for increasing response times as well as decreasing detection sensitivities.

Consequently, the TDDAN bottleneck was not fully solved. Nevertheless, according to the improvements due to the DWNCT-based ion-to-electron transducing layers, future investigations can be considered, for example, by developing a water-free, inorganic process in order to electrodeposit PEDOT:DWCNT ion-to-electron transducing layers on a platinum-based working ultramicroelectrode. This could be reached by using the acetonitrile solvent to perform EDOT electrodeposition, aiming at a long-term hydrophobicity improvement in the metal/FPSX and therefore the detection of nitrate (NO_3_^−^) ions (and other kinds of ions) in the frame of environmental analysis.

## Figures and Tables

**Figure 1 sensors-24-05994-f001:**
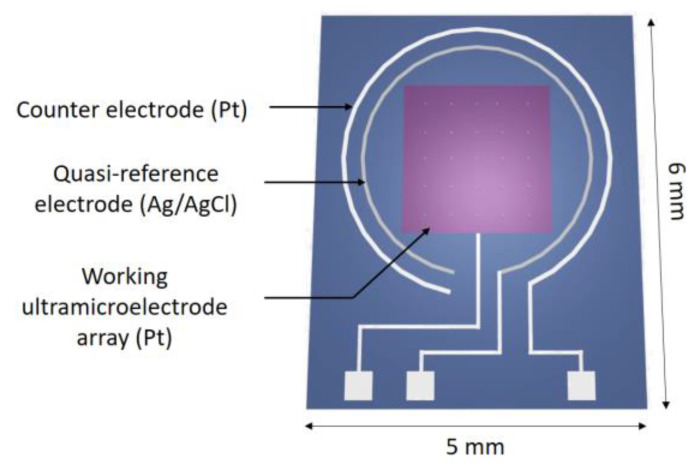
Development of silicon-based technologies for the mass fabrication of the (Pt - Pt - Pt/Ag/AgCl) electrochemical microcell (ElecCell).

**Figure 2 sensors-24-05994-f002:**
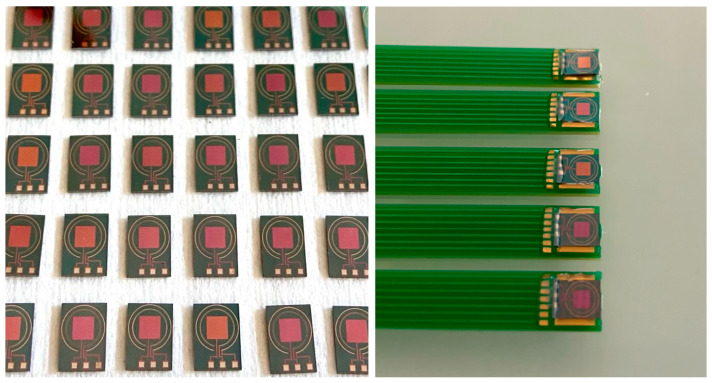
Integration of ElecCell silicon chips on printed circuit board.

**Figure 3 sensors-24-05994-f003:**
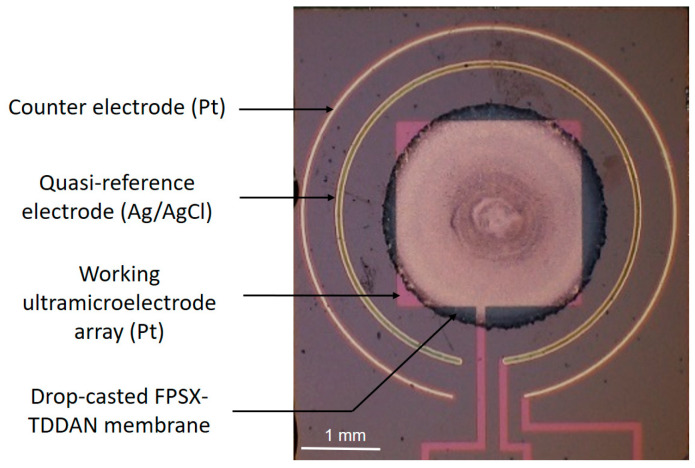
Drop-casting deposition of a fluoropolysiloxane-based nitrate ion-sensitive layer on the platinum-based modified ultramicroelectrode array used as a working electrode.

**Figure 4 sensors-24-05994-f004:**
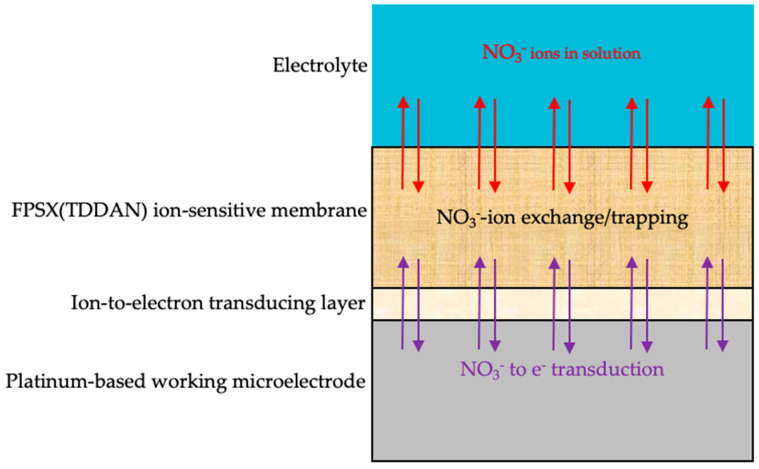
Mechanistic scheme of the Pt/IETL/FPSX(TDDAN)/electrolyte NO_3_^−^-sensitive structure.

**Figure 5 sensors-24-05994-f005:**

Optical images of the different ion-to-electron transducing layers: (**a**) PEDOT:PSS, (**b**) PEDOT:DWCNTs, (**c**) PPy:DWCNTs.

**Figure 6 sensors-24-05994-f006:**
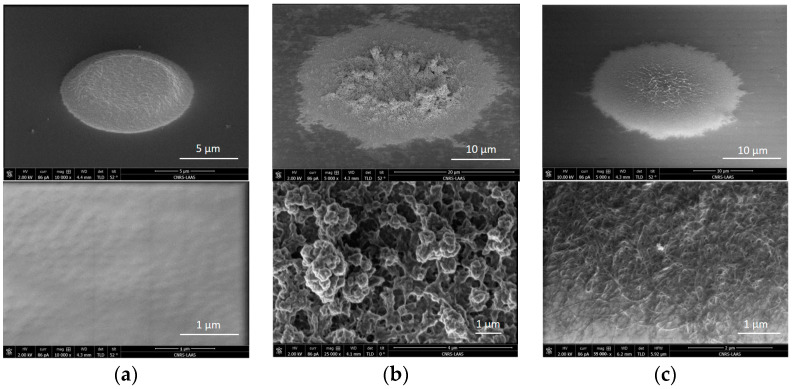
SEM images of the different IETL layers (general view (**top**) and detail (**bottom**)): (**a**) PEDOT:PSS, (**b**) PEDOT:DWCNTs, (**c**) PPy:DWCNTs.

**Figure 7 sensors-24-05994-f007:**
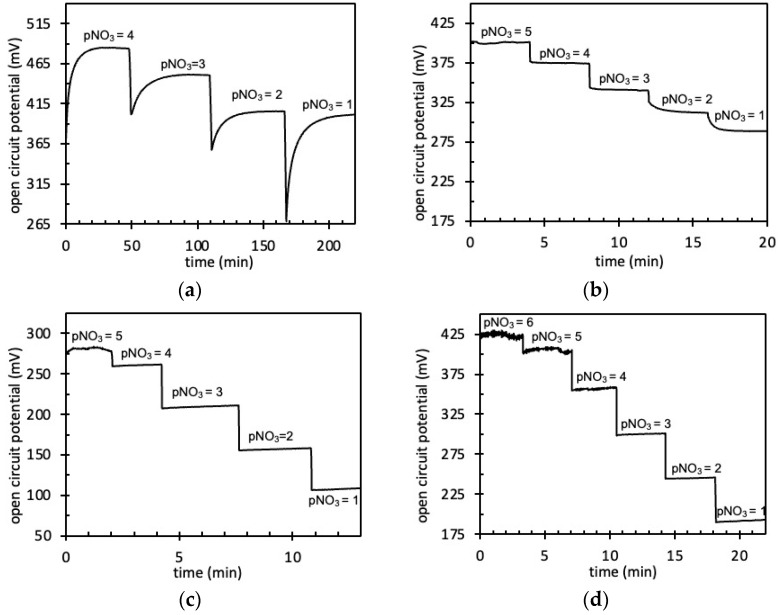
Evolution of the open-circuit potential of pNO_3_-ElecCell for different KNO_3_ solutions: (**a**) without ion-to-electron transducers, (**b**) PEDOT:PSS, (**c**) PEDOT:DWCNTs, (**d**) PPy:DWCNTs.

**Figure 8 sensors-24-05994-f008:**
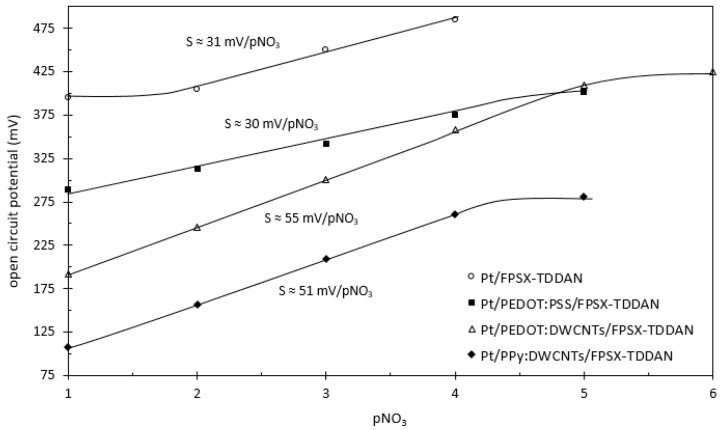
Analytical responses of the different pNO_3_-ElecCell devices.

**Figure 9 sensors-24-05994-f009:**
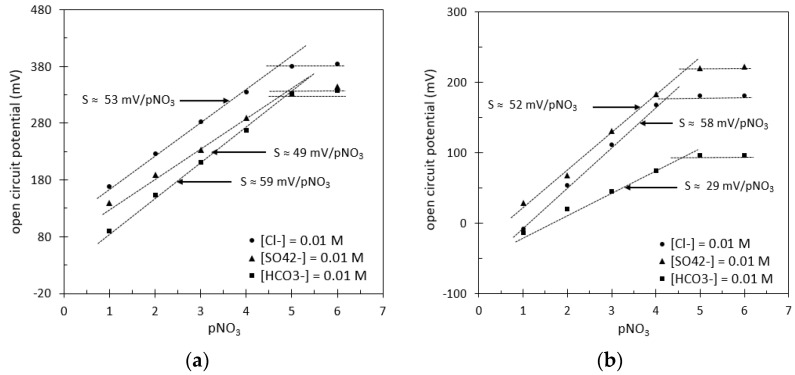
pNO_3_-ElecCell analytical response in the presence of various interfering ions: (**a**) Pt/PEDOT:DWCNT/FPSX(TDDAN) structure, (**b**) Pt/PPy:DWCNT/FPSX(TDDAN) structure.

**Table 1 sensors-24-05994-t001:** Developments and results associated with pNO_3_-ISFET-based sensors.

Reference	Insulator Material	Polymeric Membrane	Ion-Sensitive Molecule	Sensitivity (mV/pNO_3_)	pNO_3_ Linear Range
[10]	Si_3_N_4_	PVC-PVA	TDDAN	53 ± 1	from 2 to 5
[11]	Si_3_N_4_	PVA	TDDAN	53 ± 1	from 1 to 5
[12]	Si_3_N_4_	HDDA	TOAN	59 ± 1	from 1 to 5
[13]	Si_3_N_4_	PSX	TDDAN	51 ± 1	from 1 to 4
[14]	Ti	SQ	QAS	59 ± 1	from 1 to 6
[15]	AlGaN/GaN	PVC	TDDAB	40 ± 1	from 1 to 6
[16]	Si_3_N_4_	PVC-PVA	TDDAN	56 ± 1	not detailed
[17]	Si_3_N_4_	FPSX	TDDAN	56 ± 1	from 1.5 to 5.5

**Table 2 sensors-24-05994-t002:** Developments and results associated with pNO_3_-ISE-based sensors.

Reference	Electrode Material	Ion–Electron Transducer	Polymeric Membrane	Ion-Sensitive Molecule	Sensitivity (mV/pNO_3_)	pNO_3_ Linear Range
[18]	Ag	Ag/AgCl	PVC	THTDPCl	59 ± 1	from 1 to 5
[19]	Graphite	Cellulose acetate	PVC	TDDACl	54 ± 1	from 1 to 5
[20]	Glassy carbon	Graphene	PMMA	TDMAN	55 ± 1	from 2 to 4.5
[21]	Graphene	Graphene	PVC	TDMAN	55 ± 1	from 1 to 4.5
[22]	Glassy carbon	PPy	PVC	TDMAN	52 ± 1	from 1 to 5
[23]	Glassy carbon	MWCNT	PVC	Ni^+^NO_3_^−^	55 ± 1	from 1 to 5.5
[24]	Au	TRGO	PVC	NI V	59 ± 1	from 1 to 5
[25]	Glassy carbon	MWCNT composite	PVC	Co(Bphen)_2_ (NO_3_)_2_	57 ± 1	from 1 to 6
[26]	Graphene	-	PVC	TDMAN	59 ± 1	from 1 to 3.5
[27]	Graphite	-	PVC	Ag(dtc)_2_	56 ± 1	from 1 to 5

**Table 3 sensors-24-05994-t003:** Detection properties of the different pNO_3_-ElecCell devices.

Electrode Material	Ion-to-Electron Transducing Layer	Ion-Sensitive Membrane	Sensitivity (mV/pNO_3_)	pNO_3_ Linear Range	Time Constant (s)
Pt	-	FPSX(TDDAN)	31 ± 2	from 2 to 5	≈600
Pt	PEDOT:PSS	FPSX(TDDAN)	30 ± 1	from 1 to 4	≈120
Pt	PEDOT:DWCNTs	FPSX(TDDAN)	55 ± 1	from 1 to 5	<1
Pt	PPy:DWCNTs	FPSX(TDDAN)	51 ± 1	from 1 to 4	<1

**Table 4 sensors-24-05994-t004:** Temporal stability of the pNO_3_-ElecCell properties in dry storage conditions.

ISE Ion-Sensitive Structure	Day 1		Day 7		Day 30	
	Sensitivity (mV/pNO_3_)	Time Constant (s)	Sensitivity (mV/pNO_3_)	Time Constant (s)	Sensitivity (mV/pNO_3_)	Time Constant (s)
Platinum PEDOT:DWCNT FPSX(TDDAN)	55 ± 1	<1	53 ± 2	<1	42 ± 6	≈60
Platinum PPy:DWCNT FPSX(TDDAN)	51 ± 1	<1	49 ± 2	<1	40 ± 20	≈60

## Data Availability

The original contributions presented in the study are included in the article, further inquiries can be directed to the corresponding author.

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
