# Peer review of "Study of Ion-to-Electron Transducing Layers for the Detection of Nitrate Ions Using FPSX(TDDAN)-Based Ion-Sensitive Electrodes"

_sensors, 2024, doi:10.3390/s24185994_

Round 1

Reviewer 1 Report

Comments and Suggestions for Authors

The manuscript presents the development of miniaturized solid-state potentiometric sensors sensitive towards nitrate anions for environmental applications. Special attention was paid on the improvement of the response stability of the microsensors adding an ion-to-electron transducing layer between the metal and the membrane. The intermediate solid contact layer based on a conductive polymer (PEDOT:PSS) or conductive polymer/carbon nanotube composite (PEDOT or PPy/DWCNT) were applied by electrodeposition on the working electrode of the three-electrode microcell fabricated on a silicon substrate. Next, fluoropolysiloxane membrane (FPSX) doped with a well-known ion-exchanger (TDDAN) were deposited and the performances of the obtained nitrate-sensitive sensors were studied and compared. Quasi-Nernstian sensitivity, low detection limit and short response time, as well as selectivity pattern consistent with literature data for membranes with ion exchangers were achieved. Moreover, good signal stability (including low potential drift) was observed on a weekly period, but after this time the sensor properties began to deteriorate. Nevertheless, important improvement of the signal stability was obtained thanks to DWCNT-based ion-to-electron transducing layers.

In my opinion the topic of the paper is interesting and important from the point of view of fabrication of miniaturized devices dedicated to water pollution control. The proposed approach is quite simple and leads to improvement of the performance provided by proper introduction of a ion-to-electron transducing layer.

Concluding, the manuscript represents a good scientific content with reliable results. Therefore, I recommend its publication. Only a few details should be clarified to improve the scientific value of the paper, which can be published after minor revision:

1.    The Authors could compare and discuss the signal stability of the developed sensors and those reported in the literature (references 18-27).

2.    Why both anion- and cation-exchanger (in 2:1 molar ratio) were introduced into the FPSX membranes?

3.    The quality of the Figures 7 an 8 could be improved.

4.    Lines 213-215 should be removed.

Comments on the Quality of English Language

Minor editing of English language is required.

Author Response

Comment 1: The authors could compare and discuss the signal stability of the developed sensors and those reported in the literature (references 18-27).

Response 1:  There is not a lot of papers describing day-long analysis experiments, and therefore middle term stability, for nitrate-sensitive ISE (cf. references 20, 21 and 26). However, our results were compared and discussed according to these different works (cf. lines 303-309 of the revised manuscript).

Comment 2: Why both anion- and cation-exchanger (in 2:1 molar ratio) were introduced into the FPSX membranes?

Response 2: This choice of a 2:1 molar  between TDDAN and KTFBP is based on our previous experimental works dedicated to pNO3-ISFET. Nevertheless, since we are not true chemists, it was initially based on experiments/results extracted from literature. Therefore, we have no real answer to this comment.

Comment 3: The quality of the Figures 7 an 8 could be improved.

Response 3: We try to improve the figures to our best (the Sensors template being responsible for a consequent size decrease)

Comment 4: Lines 213-215 should be removed.

Response 4: Lines 2013-215 were removed (with all our apologizes for such mistake).

Reviewer 2 Report

Comments and Suggestions for Authors

The manuscript “Study of ion-to-electron transducing layers for the detection of nitrate ions using FPSX(TDDAN)-based ion-sensitive electrodes” explores the development of ion-sensitive electrodes (ISE) for detecting nitrate ions in liquid solutions. It specifically focuses on using TDDAN ion-exchanger and fluoropolysiloxane (FPSX) polymer-based layers with electrodeposited matrices containing double-walled carbon nanotubes (DWCNT). These matrices, embedded in polyethylenedioxythiophene (PEDOT) or polypyrrole (PPy) polymers, are found to enhance the ion-to-electron transduction for NO3- detection. The FPSX-based pNO3-sensors exhibit good sensitivity (up to 55 mV/pX) and selectivity against common interferents (Cl-, HCO3-, SO42-) but face challenges with temporal drift and long-term stability. The manuscript presents significant advancements in nitrate detection technology and is a valuable contribution to the field of environmental analysis. However, it needs some revisions before it can be accepted for publication.

l  The study identifies the issue of temporal drift and degradation over time. This is a critical factor for practical applications. Can the authors provide more insight into potential solutions or future research directions to address this issue.

l  In the abstract, do the authors mean “stability measurement” by saying “stable measurement”?

l  In Fig.8, please the authors pay attention to the use of subscript and superscript. For example, “SO42-” should be “SO42-”, etc.                                                                                                         

Comments on the Quality of English Language

The grammar of the draft needs to be polished.

Author Response

Comment 1: The study identifies the issue of temporal drift and degradation over time. This is a critical factor for practical applications. Can the authors provide more insight into potential solutions or future research directions to address this issue.

Response 1: We agree with reviewer comment: the issue of temporal drift and degradation over time is critical for practical use of pNO3-ISE. Apart from improving the packaging procedure, the main bottleneck is in fact related to the NO3--sensitive layers chemical properties and we have to do with TDDAN ionophore chemistry. As result, we have introduced some PEDOT:DWNCT ion-to-electron transfer layer in order to improve the long-term NO3--ion analysis. In order to go further, we plan to develop a water-free, inorganic process in order to perform EDOT polymerization (using the acetonitrile solvent) in order to further postpone the formation of a water-based layers at the metal/FPSX PEDOT-based interface. This was further explained in the revised manuscrit (see lines 358-361).

Comment 2: In the abstract, do the authors mean “stability measurement” by saying “stable measurement”?

Response 2: We agree with reviewer comment: the associated sentence was unclear and was corrected in the revise abstract (see lines 20-21).

Comment 3: In Fig.8, please the authors pay attention to the use of subscript and superscript. For example, “SO42-” should be “SO42-”, etc.

Response 3: We agree with reviewer comment. Unfortunately, the software used for the graphs realization prevent us for using subscript/superscript policies in subtitles...

Reviewer 3 Report

Comments and Suggestions for Authors

This work describes the development of ion-selective electrode (ISE) sensors for nitrate analysis in liquids. The sensors use TDDAN ion-exchanger and fluoropolysiloxane (FPSX) polymer layers, combined with electrodeposited matrices containing double-walled carbon nanotubes in PEDOT or PPy polymers. These improved the ion-to-electron transduction for NO3- detection. The resulting FPSX-based pNO3-ElecCell microsensors showed good sensitivity and acceptable selectivity against common interfering anions. The idea is interesting, and authors put a lot effort into their research. However, there are some minor issues, authors need to pay attention on before Manuscript can be published.

1.     Abstract: Please, explain the abbreviation for TDDN as well as in other cases.

2.     The Introduction typically doesn't include tables. Consider moving the comparative data to the Results and Discussion section, dedicating a paragraph to contrasting your findings with existing literature. This comparison is more pertinent there. In the Introduction, a brief mention of previous work and approaches in the field would suffice.

3.     “This section may be divided by subheadings. It should provide a concise and precise description of the experimental results, their interpretation, as well as the experimental conclusions that can be drawn.” This explanation is not needed.

4.     Please, emphasize the novelty and significance of the presented work more throughout the Manuscript, especially in the Conclusions.

Author Response

Comment 1: Abstract: Please, explain the abbreviation for TDDAN as well as in other cases.

Response 1: the TDDAN abbreviation was explained in the revised abstract (see lines 13).

Comment 2: The Introduction typically doesn't include tables. Consider moving the comparative data to the Results and Discussion section, dedicating a paragraph to contrasting your findings with existing literature. This comparison is more pertinent there. In the Introduction, a brief mention of previous work and approaches in the field would suffice.

Response 2: In the introduction, we have perform a study of the different pNO3 sensors developed in the literature, either based on ISFET and ISE technologies. The objective is to emphasize that many different polymers, ionophores, ion-exchangers or ion-to-electron transferring layers were studied in order to cope with the NO3--ion long-term detection, allowing us to present our works accordingly. It seems to us that such approach fit with a typical Introduction and we hope to convince the reviewer in this way.

Comment 3: “This section may be divided by subheadings. It should provide a concise and precise description of the experimental results, their interpretation, as well as the experimental conclusions that can be drawn.” This explanation is not needed.

Response 3: This section was removed (with all our apologizes for such mistake).

Comment 4: Please, emphasize the novelty and significance of the presented work more throughout the Manuscript, especially in the Conclusions.

Response 4: the conclusion was revised according to reviewer comment (see lines 349-361).

Reviewer 4 Report

Comments and Suggestions for Authors

Camille Bene et al., reported development of an ISE based nitrate sensor. The article is well written and informative. I have several minor suggestions for improvement.

1) Why this type of sensor outdates other electrochemical sensors as discussed in DOI: 10.1016/j.electacta.2020.135994  

2) Please provide a mechanistic scheme of the sensor developed.

3) In the mechanistic scheme explain the roles of different interfering components. 

4) Please properly discuss reusability and stability.

5) Please check English. There are several incongruities  

Comments on the Quality of English Language

Minor revisions

Author Response

Comment 1: Why this type of sensor outdates other electrochemical sensors as discussed in DOI: 10.1016/j.electacta.2020.135994 ?

Response 1: Nitrate analysis in liquid phase can be performed according to two detection techniques, either by amperometry using the  oxidation/reduction properties of the NO3-/NO2- chemical system, either by potentiometry using the electrical valence of the NO3- ion. In the proposed manuscrit, we focused on the second one and it seems to us difficult to compare both as requested by the reviewer (in fact, such works should be associated to a complete review article on NO3- ion detection...). We hope that the reviewer will understand such position.

Comment 2: Please provide a mechanistic scheme of the sensor developed.

Comment 3: In the mechanistic scheme explain the roles of different interfering components.

Responses 2 and 3: A new figure was added in the revised paper as suggested (see figure 4).

Comment 4: Please properly discuss reusability and stability.

Response 4: Reusability and stability were discussed in the revised manuscript (cf. lines 303-317).

Comment 5: Please check English. There are several incongruities.

Response 5: English was checked and corrected to our best.